# The Development of Molecular Biology of Osteoporosis

**DOI:** 10.3390/ijms22158182

**Published:** 2021-07-30

**Authors:** Yongguang Gao, Suryaji Patil, Jingxian Jia

**Affiliations:** 1Tangshan Key Laboratory of Green Speciality Chemicals, Department of Chemistry, Tangshan Normal University, Tangshan 063000, China; jjx1027@163.com; 2Lab for Bone Metabolism, School of Life Sciences, Northwestern Polytechnical University, Xi’an 710072, China; suryajip@mail.nwpu.edu.cn

**Keywords:** osteoporosis, bone remodeling, osteoblasts, osteocytes, osteoclasts, bone therapeutics

## Abstract

Osteoporosis is one of the major bone disorders that affects both women and men, and causes bone deterioration and bone strength. Bone remodeling maintains bone mass and mineral homeostasis through the balanced action of osteoblasts and osteoclasts, which are responsible for bone formation and bone resorption, respectively. The imbalance in bone remodeling is known to be the main cause of osteoporosis. The imbalance can be the result of the action of various molecules produced by one bone cell that acts on other bone cells and influence cell activity. The understanding of the effect of these molecules on bone can help identify new targets and therapeutics to prevent and treat bone disorders. In this article, we have focused on molecules that are produced by osteoblasts, osteocytes, and osteoclasts and their mechanism of action on these cells. We have also summarized the different pharmacological osteoporosis treatments that target different molecular aspects of these bone cells to minimize osteoporosis.

## 1. Introduction

Osteoporosis is a well-known skeletal disease that is characterized by declined bone strength due to weakened bone microarchitecture and increased vulnerability to fractures [1]. Osteoporosis is a prominent cause of concern in the aging population, and the intensity of osteoporotic fractures becomes prevalent with age and becomes challenging. As a result, it affects the physical, psycho-social, and financial aspects of human life [2]. Osteoporosis and osteoporosis-related fractures are predominant in older adults and post-menopausal women, leading to morbidity. Post-menopausal women are predominantly disposed to suffering from common bone fractures such as vertebral and hip fractures [3,4]. Osteoporosis is classified into two types: primary and secondary osteoporosis, with up to 30% of post-menopausal women and 50 to 80% of men having factors contributing to osteoporosis. When osteoporosis is age-related and arises in women post-menopause and men in the absence of an underlying disease, it is called primary osteoporosis. When it occurs in the presence of an underlying disease or medication, it is called secondary osteoporosis [5].

The most common methods of diagnosis include the measurement of bone mineral density (BMD) using dual-energy X-ray absorptiometry (DEXA) and quantitative computed tomography (QCT) [6], and the fracture risk calculation by using the Garvan fracture risk calculator, QFracture^®^, and fracture risk assessment tool (FRAX^®^) [7]. The available treatment guidelines provide recommendations on when to treat patients but do not specify the types of drugs to prescribe in various situations. Osteoporosis therapy needs to be based on individuals, and needs to consider the dosing regimen to achieve desired efficacy, safety, and cost [8]. As BMD, osteoporosis, and osteoporotic fractures are extremely heritable, understanding the molecular mechanisms of osteoporosis is important to develop effective and efficient therapeutics to reduce and treat osteoporosis. Bone remodeling is a coordinated process involving bone-forming osteoblasts and bone-resorbing osteoclasts to maintain bone mass, repair micro-damage, and mineral homeostasis [9]. Osteoporosis signifies an imbalance in the bone remodeling process because of the excessive osteoclast-mediated bone resorption or inadequate osteoblast-mediated bone formation. This leads to a loss in both trabecular and cortical bones, causing spinal and hip fractures, and long bone fractures, respectively. To sustain bone homeostasis, several molecules harmonizes osteoblast and osteoclast functions [10]. Many hormones, especially estrogens, are known to regulate bone remodeling by controlling cytokines and growth factors produced from bone marrow as well as bone cells [11]. In the present review, we have summarized the cellular and molecular mechanisms that control bone remodeling through osteoblasts, osteocytes, and osteoclasts and possible targets for osteoporosis treatment. Additionally, we have highlighted the anti-resorptive and anabolic osteoporosis treatment.

## 2. Bone Remodeling

Bone modeling is the process in which osteoclasts and osteoblasts work independently to degrade old bone and form a new one, respectively. This process is important in defining the progress of osteoporosis. Bone remodeling is crucial for adult bone homeostasis and involves balanced coordination in bone formation and resorption to support bone mass and systemic mineral homeostasis. Bone remodeling can be divided into five phases: activation phase, resorption phase, reversal phase, formation phase, and termination phase [12,13]. These steps take place simultaneously but asynchronously at multiple locations within the skeleton.

The significant signaling pathways that control osteoclastic bone resorption and osteoblastic bone formation include receptor activator of nuclear factor-κB (RANK)/RANK ligand (RANKL)/osteoprotegerin (OPG) and canonical Wnt signaling (Figure 1). The process is also regulated by paracrine regulators such as cytokines, growth factors, prostaglandins, and endocrine regulators such as parathyroid hormone (PTH), calcitonin, vitamin D, glucocorticoids, and growth hormone, and sex hormones [14]. The activation phase involves recruitment and activated osteoclast precursor cells and bone surface exposure by separating bone lining cells from the bone. This separation creates a canopy termed bone-remodeling compartment (BRC) over this site. Multiple mononuclear cells fuse to generate multinucleated preosteoclasts, forming sealing zones by binding to the bone matrix around bone-resorbing compartments, isolating the resorption pit from surrounding bone. Osteocytes provide initiating signal through their network of dendritic processes to signal other cells. Apoptosis of osteocytes also releases paracrine factors that increase the recruitment of osteoclast and osteoblast precursors [14]. Local as well as systemic regulators promote osteoclastogenesis. Transforming growth factor-beta (TGF-*β*), macrophage colony-stimulating factor (M-CSF) and RANKL are constituted as local regulators, whereas vitamin D, calcium, PTH, and estrogen act as systematic regulators [12]. Osteoclast attachment to bone stimulates changes into the cytoskeleton leading to the formation of a filamentous actin ring or sealing zone, which surrounds the ruffled border. This forms a separate resorptive environment between the osteoclast and the underlying bone matrix. The ruffled border is produced because of the fusion of secretory vesicles with the bone-apposing plasma membrane. A calcium sensor protein, Synaptotagmin VII is essential for the formation of the ruffled border in osteoclasts [15]. 

Bone resorption is determined by the formation and stability of the sealing zone, which is constituted of actin and integrin-based podosomes. Actin ring formation in the sealing zone consists of contractile proteins and is stabilized by actin and tubulin microtubules [16]. Vacuolar ATPases (V-ATPases) interaction with actin via the N-terminal domain and accumulation at the ruffled membrane leads to the release of H^+^ into resorption lacuna. The acid and proteases, such as matrix metalloproteinases (MMPs), dissolve crystalline hydroxyapatite and the organic matrix, respectively [17].

After bone resorption, the osteogenic environment is prepared at remodeling sites for bone formation by osteoclasts by releasing growth factors from the resorbed matrix, producing secreted proteins, and expressing membrane-bound factors that facilitate bone formation [18]. Once mesenchymal stem cells or osteoblast progenitors are returned to the resorption lacunae, they differentiate and secrete collagen type I, a primary component of bone. Non-collagenous proteins, such as proteoglycans, alkaline phosphatase, small integrin-binding ligand (SIBLING) proteins, Gla-containing proteins (matrix Gla protein and osteocalcin), and lipids then form the remaining organic material and hydroxylapatite is deposited into new bone, replacing the resorbed bone [19]. When bone loss is compensated, osteoblasts terminally differentiate into osteocytes and bone remodeling is terminated.

## 3. Molecular Biology of Osteoporosis

### 3.1. Osteoblasts

Due to the rapid developments in cellular metabolism research, many investigators have focused on the understanding of the pathophysiology of chronic diseases to investigate novel therapeutic targets in bone cells such as osteoblasts. Osteoblasts, derived from undifferentiated mesenchymal cells, are essential bone-forming cells required for the growth and maintenance of the skeleton, are regulated by various molecules (Figure 2). Dysfunction in osteoblast due to disorders of substrate availability such as diabetes mellitus, anorexia nervosa, and aging eventually leads to skeletal fragility and osteoporotic fractures [20].

#### 3.1.1. Wnt Signaling

The Wnt-*β*-catenin signaling pathway is an evolutionarily conserved system of cell-cell communication that plays an important role in stem cell renewal, cell proliferation, and cell differentiation. The Wnt signaling pathway can be categorized into two pathways: canonical *β*-catenin-dependent pathway and non-canonical *β*-catenin-independent pathways. Genetic or epigenetic changes that might lower or over-activate the Wnt-*β*-catenin signaling can increase human diseases, including osteoporosis [21]. Wnt proteins are the members of the secreted glycoproteins family and play critical roles during osteoblast differentiation. Their signals are transduced by a family of seven-pass transmembrane G-protein coupled receptors of the frizzled (Fzd) family, and a co-receptor of the low-density lipoprotein receptor-related protein (LRP) family (e.g., LRP5 and LRP6), or a Ryk or Ror transmembrane tyrosine kinase [22].

Wnt signaling regulates mesenchymal stem cells (MSC) differentiation into osteoblasts and inhibits adipogenesis by preventing the expression of the adipogenic transcription factors C/EBPα and PPARγ [23]. Wnt family members like Wnt10a, Wnt10b, and Wnt6 have been found to regulate the fate of MSC. The gain- and loss-of-function study showed how Wnt6, Wnt10a, and Wnt10b regulate mesenchymal cell fate. The overexpression of Wnt10a and Wnt10b greatly stabilizes *β*-catenin, suppresses adipogenesis, and stimulates osteoblastogenesis through a canonical *β*-catenin-dependent pathway, whereas Wnt6 has a weaker effect. However, knockdown of Wnt6 enhanced preadipocyte differentiation and weakened osteoblastogenesis compared to Wnt10a or Wnt10b [24]. LRP5 expressed by osteoblasts, encodes the low-density lipoprotein receptor-related protein 5, a co-receptor required for signal transduction that affects bone mass. The mutations in LRP5 can lead to osteoporosis-pseudoglioma syndrome (OPPG), exhibiting reduced bone mass [25]

Wnt Signaling is checked by the inhibitors such as secreted frizzled-related protein (sFRP), Dickkopf-1 (DKK1), and sclerostin. sFRP acts as a decoy receptor of Wnt and inhibits the binding of the ligand to the receptor complex and canonical as well as non-canonical Wnt signaling pathways. While DKK1 binds to the BP1 and BP3 domains of the LRP5/6 receptor and formulate a complex with Kremens to increase internalization of the LRP5/6 receptor and inhibits the canonical Wnt pathway, sclerostin suppresses bone formation by obstructing the canonical Wnt signaling pathway. Sclerostin binds to the BP1 domain of the LRP5/6 receptor as well as LRP4 to enhance its inhibitory effects on the canonical Wnt signaling pathway [26]. The WNT signaling pathway has been identified as a candidate for therapeutic intervention to increase bone mass and strength. Particularly, the inhibition of sclerostin has been confirmed to have a very efficient osteo-anabolic effect increasing bone formation and bone density and reducing fracture risk in treated patients [27].

#### 3.1.2. Runt-Related Transcription Factor 2 (RUNX2)

RUNX2 is an upstream transcription factor involved in the regulation of osteoblast differentiation, chondrocyte maturation, as well as osteoclastogenesis. In osteoblasts, RUNX2 regulates the expression of Sp7 protein, which is a key transcription factor for osteoblast differentiation and bone matrix genes [28]. In pre-confluent G(0)/G(1) transition immature MC3T3 osteoblastic cells, RUNX2 levels were upregulated with maximum expression levels during early G(1). Interestingly, enforced expression of RUNX2 suppressed proliferation of MC3T3 preosteoblasts and C2C12 mesenchymal cells by delaying G(1) phase [29]. 

However, the reestablishment of RUNX2 into RUNX2^−/−^ calvarial osteoblast cells restored cell growth control and regulated normal osteoblast proliferation [30]. Fibroblast growth factor receptor (FGFR) genes that are expressed in mesenchymal cells constitute the important FGF signaling to control proliferation, anti-apoptosis, angiogenesis, epithelial-to-mesenchymal transition, and crosstalk with the canonical Wnt signaling. RUNX2 directly regulates FGFR2 and FGFR3 to increase the proliferation of osteoblast progenitors [31]. While RUNX2 also controls the commitment of mesenchymal cells to the osteoblast lineage by enhancing the expression of Col1a1 and Bglap and activating their promoters, RUNX2 overexpression inhibits osteoblast maturation and decreases Col1a1 and Bglap expression leading to the arrest of osteoblasts in an immature stage [32]. The change in the expression or activity of RUNX2 affects osteoblast differentiation and RUNX2 can be regulated by post-translational modifications such as phosphorylation, ubiquitination, and acetylation [33]. The upregulated RUNX2 binds to the DNA element in the Osx promoter and activates promoter of Osterix (Osx) in mesenchymal cells to regulate bone and cartilage formation [34].

#### 3.1.3. Osteocalcin

Osteocalcin, also known as bone γ-carboxyglutamic acid-containing protein (BGLAP) is a non-collagenous, vitamin K-dependent protein secreted by osteoblasts. The presence of the three residues of γ-carboxyglutamic acid allows osteocalcin to bind to calcium and subsequently hydroxyapatite. The levels of carboxylated and undercarboxylated forms of osteocalcin are often used in clinical practice and research as a marker of bone turnover to predict fracture or diabetes risks [35,36]. Osteocalcin maintains the structural properties of bone by interacting with hydroxyapatite and collagen I. The levels of glycation are associated with bone fragility and therefore, the addition of sugar on a protein can affect osteocalcin’s ability to interact with collagen I or osteopontin (OPN) but not with hydroxyapatite [37]. The knockdown of osteocalcin in MSCs delays the maturation of mineral species and lowers total hydroxyapatite. 

Moreover, the expression of RUNX2, alkaline phosphatase, type I collagen, and osteonectin genes was reduced during osteogenic differentiation of MSC [38]. Osteocalcin promotes the absorption of calcium into the hydroxyapatite and leads to bone mineralization. The reduction in bone mineralization frees osteocalcin and becomes circulated in the blood [39]. Osteoporosis is described to have decreased hydroxyapatite crystal formation and therefore results in elevated levels of osteocalcin in serum. Hence, serum osteocalcin can be employed as a diagnostic biomarker to screen primary osteoporosis in women [40].

#### 3.1.4. RANKL

RANKL is a type II transmembrane protein produced by the osteoblasts and other stromal cells with an extracellular domain at the C-terminus. [41]. It is present as a membrane-bound and soluble form, which is produced by the proteolytic cleavage of the ectodomain by metalloproteinases and released to the extracellular environment. Both forms of RANKL act as agonistic ligands for RANK, but the membrane-bound RANKL is more effective than the soluble form [42]. RANKL exerts its osteoclastic effect by binding to its corresponding receptor RANK to promote RANK signaling and subsequently osteoclastogenesis [43]. Mice deficient in soluble RANKL have no effect on bone mass or structure in growing mice but decreases osteoclast number and stimulates cancellous bone mass in adult mice. Additionally, estrogen deficiency-induced bone loss also remained unaffected by the absence of soluble RANKL. This suggests that the membrane-bound RANKL is adequate for physiological bone remodeling in adult mice [44]. RANKL addition in cultured RAW264.7 cells demonstrated increased bone resorption pit, F-actin, and levels of osteoclastogenesis specific marker genes, which suggested amplified differentiation of RAW264.7 cells into multinucleated osteoclastic cells [45]. Mice lacking RANKL in osteocytes show reduced osteoclast number and the bone loss resulted due to ovariectomy and no surge in bone marrow B lymphocytes caused by ovariectomy. This demonstrates that RANKL produced by osteocytes is responsible for the increase in B cells and bone loss that are resulted from estrogen deficiency in mice [46].

#### 3.1.5. OPG

OPG, a soluble decoy receptor for RANKL is mainly produced by osteoblasts. OPG inhibits osteoclast formation and bone resorption by inhibiting the interaction of RANKL-RANK [47]. The deficiency of OPG (OPG^−/−^) induces osteoporosis in mice due to increased bone resorption but also enhances bone formation. The application of bone resorption inhibitor risedronate in OPG^−/−^ mice significantly decreased mineral apposition rate and osteoblast surface/bone surface as well as lowered high serum alkaline phosphatase activity and osteocalcin concentration. Bone morphogenetic protein-2 (BMP-2) establishment into OPG^−/−^ mice did not accelerate ectopic bone formation even with a high turnover rate of bone; however, ectopic bone mineral density reduction was higher. This indicates bone formation and bone resorption are coupled at local sites in OPG^−/−^ mice [48]. According to Koide, et al., OPG^−/−^ mice also had higher alveolar bone loss and severe bone resorption in cortical areas of the alveolar bone due to increased osteoclasts number in the area [49]. Alveolar bone loss can be restored by the treatment of WP9QY (W9) peptide, a RANKL blocker, in OPG^−/−^ mice, which suppress osteoclastogenesis and increase osteoblastogenesis. The treatment of W9 in OPG^−/−^ mice decreased the osteoclast number in the alveolar bone and enhanced Wnt/*β*-catenin signaling to induce alveolar bone formation in OPG^−/−^ mice. OPG^−/−^ mice also express sclerostin in a low amount in tibiae and W9 treatment further repressed sclerostin expression [50]. The site of OPG production also matters in regulating bone and immune homeostasis. In OPG-floxed mice, it showed that locally produced OPG is important for bone and immune homeostasis rather than circulating OPG. It suggests that OPG limits its role within the tissue where it was produced [51].

#### 3.1.6. Osx

Osx, also known as transcription factor SP7 is a zinc finger-containing protein that belongs to the Sp/KLF (Kruppel like Factor) factors [52]. The importance of Osx in osteoblast differentiation and bone formation was demonstrated in Osx null mice. In these mice, mesenchymal cells could not deposit bone matrix, whereas cells in the periosteum failed to differentiate into osteoblasts [53]. In adult bone, inactivation of Osx in all osteoblasts reduced BMD and bone-forming rate in a lumbar vertebra, and reduced the length of the long bones making cortical bone thinner and more porous but increased immature trabecular bones [54]. Moreover, the inactivation of Osx in already formed bones produces a functional defect in osteoblasts reducing bone formation, though does not affect osteoblast proliferation or osteoclast formation [55]. Lee, et al. have shown that Osx targets the different genes and mechanisms that regulate osteoblast differentiation by knocking down Osx in MC3T3-E1 osteoblastic cells. Osteoblasts displayed a significantly reduced cell differentiation and nodule formation, and Osx-knockdown upregulated 15 and downregulated two genes associated with cell differentiation. 

Of the 15 upregulated genes, the expression of fibrillin-2 and periostin was considerably higher, suggesting the possibility of fibrillin-2 and periostin as target candidates of Osx in osteoblast differentiation [56]. Osx also contributes to corticalization for longitudinal bone growth in long bones. In osteoblast-specific Osx-knockout mice (Col-OM), corticalization was delayed, and femoral length was decreased. Mice also displayed compromised matrix coalescence and osteoblast migration. Integrin-mediated focal adhesion composed of vinculin, which mediates cell adhesion, migration, and signaling, was also reduced in osteoblasts from Col-OMT mice. The protein levels of integrin *β*3 were diminished in Col-OMT mice [57]. Among the various gap junction protein in bone cells, connexin43 (Cx43) is the most abundant and contributes to osteoblast differentiation. The study demonstrates a proportional relationship between Osx and Cx43. When Osx was silenced using shRNA, Cx43 expression was also suppressed. By direct occupying the promoter region of Cx43, Osx subsequently improved bone morphogenetic protein 4-induced Cx43 promoter activity [58]. Osx also affects hedgehog (Hh) signaling during bone development. Knockdown of Osx expression in MC3T3-E1 cells completely blocks Sonic hedgehog-induced osteoblast differentiation indicating Osx is important in early osteoblast differentiation [59].

#### 3.1.7. Core-Binding Factor Subunit Beta (CBFB)

Cbfa1, also known as Pebp2αA (polyoma enhancer-binding protein), is a transcription factor that belongs to the runt-domain gene family (Cbfa1/Pebp2αA, Cbfa2/Pebp2αB, and Cbfa3/Pebp2αC) [60]. Cbfa1 is expressed exclusively in a cell type that develops into both osteoblast and chondrocyte but later during development, its expression is limited to osteoblast progenitors and differentiated osteoblasts [61]. Cbfa1 promotes the development of osteoblasts and targeted deletion of Cbfa1 prevents osteoblasts and osteoclasts maturation resulting in a complete lack of bone formation [60]. The Cbfb deleted mice (Cbfb^fl/fl/Cre^ mice) displayed dwarfism, stunted ossifications, and osteoblast differentiation. The molecular effect of Cbfb deletion in Cbfb^fl/fl/Cre^ primary osteoblasts was that the protein levels of Runx1, RUNX2, and Runx3 were reduced and affected the stability of RUNX2. Thus, Cbfb modulates bone development by stabilizing Runx family proteins [62].

#### 3.1.8. BMP-2

BMPs are growth factors that belong to the TGF-*β* superfamily. Over 20 BMPs have currently been identified and play vital roles in cardiogenesis, neurogenesis, eye formation, as well as osteogenesis, adipogenesis, chondrogenesis. BMP-2 is the first BMP that was identified in bone and since then, its functions in bone have been deeply elucidated in development and osteogenesis [63]. BMPs are secreted as cytokines that signal through three transmembrane type II receptors and four transmembrane type I receptors. The binding of BMP promotes phosphorylation of the type I receptor by the type II receptor and subsequently activates intracellular signaling, which is initiated by phosphorylation of receptor-regulated SMADs (R-SMADs). The formation of heteromeric complexes by activated R-SMADs with SMAD4 triggers transcriptional responses [64].

As bone fractures are common in the population with osteoporosis, BMPs have been suggested to have therapeutic potential to improve fracture healing in such patients [65]. It is known that mechanical stimulation enhances bone formation, and mechanical loading together with BMP-2 regulates early signaling events in the BMP pathway. Mechanical and BMP-2 stimulation increases phosphorylation of R-Smads and maintains its phosphorylation and regulates osteogenesis [66]. BMP-2 induces canonical Smad signaling and non-canonical pathways by activating MAPK cascades through Smad. BMP-2 stimulation upregulates the expression of multiple types of αv*β*-integrins in human osteoblasts, which in turn mediates Smad signaling via the Cdc4-Src-FAK-ILK cascade. These interactions of BMP-2 signaling and mechanotransduction ultimately leads to the translocation of YAP/TAZ and activate osteogenic genes and drives osteogenic differentiation. 

Moreover, in the absence of BMP-2, the Smad complexes remain bound and active on target genes [67]. The study conducted in osteoporosis rats showed that the overexpression of BMP-2 in rat bone marrow mesenchymal stem cells (rBMSCs) activates the BMP signaling pathway promoting osteoblastic differentiation, while the low expression of BMP-2 suppresses the BMP signaling pathway in rBMSCs [68]. The addition of BMP-2 in osteoporosis rats significantly improved bone repair and enhanced new bone formation in the margins of the defect as well as intramembranous ossification zones by reducing mineralization of the newly formed bone [69].

#### 3.1.9. Forkhead Box Class O Family Member Proteins (FoxOs)

FoxOs are transcription factors that convert environmental signaling into gene expression to regulate cell survival, proliferation, differentiation, and oxidative stress resistance. It has been reported that loss- or gain-of-function of FoxOs can significantly alter the bone cell function and intercellular signaling, and dysfunction can lead to osteoporosis or other bone diseases [70]. FoxO activities in osteoblast significantly impact bone formation. In osteoblast progenitors, FoxOs inhibit Wnt signaling and bone formation, while increases osteoprotegerin expression in cells of the osteoblast lineage and decrease bone resorption by osteoclasts [71]. The previous study has demonstrated that by deleting FoxO1, 3, and 4 in osteoblast progenitor; cancellous bone mass loss can be mitigated in a mouse model of type 1 diabetes [72]. Under the influence of osteogenic stimulants, mouse mesenchymal cells exhibit increased FoxO1 expression and activity. This increased expression stimulates the expression of osteogenic markers such as RUNX2, alkaline phosphatase, and osteocalcin to promote mesenchymal cell differentiation into osteoblasts [73]. Additionally, FoxO1 acts as a negative regulator of RUNX2, osteoblast-specific transcription factor, and controls IGF1/insulin-dependent regulation of osteocalcin expression in osteoblasts [74]. The study has shown that downregulation of FoxO1, reduces mRNA levels of RUNX2, type 1 collagen, OCN, and MMP13, as well as the formation of mineralized nodules. However, overexpression of FoxO1 reduces the proliferation of MC3T3-E1 cells [75].

#### 3.1.10. Nuclear Factor E2 p45-Related Factor 2 (Nrf2)

Nrf2 is an essential transcription factor required for the regulation of detoxifying and antioxidative genes and is an important modulator of carbohydrate, lipid, heme, and iron metabolisms. Nrf2 promotes survival of irradiated osteoblasts in hematopoietic compartments Nrf2 to support long-term hematopoietic stem cell niches [76]. In osteoblasts of cancellous bone in the femur of ovariectomized (OVX) mice, high expression of Nrf2 mRNA and protein was detected. In vitro, the nrf2 expression vector inhibited an increase in alkaline phosphatase activity and the mineralized matrix formation in MC3T3-E1 cells because of the impaired RUNX2-dependent stimulation of osteocalcin promoter activity. The recruitment of RUNX2 on osteocalcin promoter was also impaired without affecting the expression of RUNX2 mRNA. This indicates that Nrf2 acts as a negative regulator of cellular differentiation by inhibiting RUNX2-dependent transcriptional activity in osteoblasts [77].

### 3.2. Osteocytes

Osteocytes are mature osteoblasts that have a more differentiated morphology and, in time, become embedded in the matrix that they are actively synthesizing during the process of bone formation. Osteocytes maintain contact with adjacent osteocytes, osteoblasts, and endothelial cells and communicate via gap junctions [78]. The osteocyte cell and its dendritic processes are present in a complex lacuno-canalicular system and are important mechanosensing cells in bone. Osteocytes translate mechanical force on the bone to cellular signaling to maintain bone homeostasis. Various proteins and signaling molecules such as sclerostin, cathepsin K, Wnts, DKK1, DMP1, IGF1, and RANKL/OPG are produced by osteocytes to regulate osteoblast and osteoclast activity [79]. An altered expression of these molecules can lead to certain pathogenic conditions, including osteoporosis. Therefore, it is important to understand their function in the bone.

#### 3.2.1. Stimulatory Subunit of G-Protein (Gsα)

In osteocytes, the stimulatory subunit of G-protein (Gsα) acts as a secondary messenger of G protein-coupled receptors (GPCRs). The absence of Gsα in osteocytes in mice induces high expression of sclerostin, which suppresses osteoblast activity, and decreases trabecular and cortical bone. Moreover, the deletion of Gsα also affects osteocyte morphology. Mice lacking GsαKO were deficient in dendrites and were randomly distributed throughout the bone matrix [80]. Additionally, deficiency of Gsα in osteocytes alters myelopoiesis and osteopenia. Osteocytes secrete Gsα-dependent factor(s), neuropilin-1 (Nrp-1), and granulin (Grn), and regulate myeloid cells proliferation by promoting myeloid and macrophage cells proliferation and differentiation into bone-resorbing osteoclasts [81].

#### 3.2.2. Sclerostin

Sclerostin is highly expressed and secreted by osteocytes. It binds to LRP 5/6 and inhibits Wnt signaling, suppressing osteoblast differentiation [82,83]. The increased levels of circulating sclerostin in humans as well as in animal models have been reported during immobilization and prolonged bed rest. The mechanical unloading showed that it increases SOST/sclerostin expression in osteocytes [84]. Sclerostin regulates the expression of osteocyte-specific proteins, such as RANKL, OPG, and proteins encoded by DMP1, PHEX, and possibly FGF23. The unloading increases sclerostin levels which antagonize Wnt-canonical-*β*-catenin signaling pathways in osteocytes and osteoclasts, while mechanical loading lowers sclerostin levels and activates Wnt-canonical signaling and bone formation [85].

#### 3.2.3. Integrins

Integrins in osteocytes are required for cell adhesion and signaling and are vital in facilitating mechanotransduction and, therefore, in load-induced bone formation. Osteocytes express *β*1 and *β*3 integrins subunits, and antagonism of these integrins result in osteoporosis, osteoarthritis, and bone metastases [86]. Osteocyte processes are more sensitive to mechanical loading than their cell bodies. The study showed that the membrane proteins involved in osteocyte mechanotransduction, such as purinergic channel pannexin1, the ATP-gated purinergic receptor P2 × 7R, and the low voltage transiently opened T-type calcium channel CaV3.2-1 all were preferentially localized close to *β*3 integrin attachment foci on osteocyte processes [87]. The integrin αv*β*3 is important in maintaining osteocyte morphology and for mechanosensation and mechanotransduction. The integrin αv*β*3 antagonism has shown that it alters osteocyte cell processes, reduces spread area, and process retractions. Moreover, in response to fluid shear stress, the blocking of integrin αv*β*3 also interrupts COX-2 expression and PGE2 release [88].

### 3.3. Osteoclasts

Osteoclasts are the primary bone-resorbing cells that contribute to the remodeling of the bone and therefore, defects in osteoclasts can result in unbalanced bone remodeling, causing pathological disorders such as bone metastasis, osteoporosis, and inflammatory bone erosion [89]. These are multinucleated giant cells produced by fusion of hematopoietic stem cells-derived monocytic precursors in the presence of colony-stimulating factor 1 (CSF1) and RANKL [90]. Osteoclasts remove damaged bone during the remodeling of the bone to maintain a healthy skeleton by releasing acid into the resorption lacunae. Several cytokines such as PU.1, M-CSF, c-Fos, CCAAT/enhancer-binding protein α (C/EBPα), RANK, nuclear factor of activated T-cells cytoplasmic 1 (NFATc1), and microphthalmia-associated transcription factor (MITF) regulates osteoclastogenesis and osteoclast differentiation and by regulating the expression of RANKL and its inhibitory decoy receptor OPG [91,92] (Figure 3).

PU.1 is a predominantly expressed member of the Ets family protein. PU.1 is involved in promoting the expression of NFATc1 and binding to the NFATc1 promoter to increase osteoclast-specific gene expression [93]. During bone marrow-derived macrophages (BMMs) differentiation into osteoclast, NFATc1 and interferon regulatory factor (IRF) 8 play positive and negative roles, respectively. The study has shown that PU.1, during RANKL-induced osteoclastogenesis, favors NFATc1 over IRF8 and modifies the binding regions that are associated with changes in epigenetic profiles, and thus controls cell type-specific gene expression [94]. Moreover, PU.1, in association with MITF coordinates a transcription factor network essential for osteoclast differentiation. Many of those are regulated by the binding of PU.1 with BRD4, forming superenhancers. The chromatin conformational changes in the superenhancer region of NFATc1 are also dependent on PU.1 [95].

MITF is essential for terminal osteoclast differentiation and shift from the cytoplasm to the nucleus upon RANKL/CSF-1 action. 14-3-3 chaperone-like adaptor proteins engage different proteins and regulate DNA replication, cell proliferation, differentiation, and apoptosis, which also is a binding partner of MITF in osteoclast precursors. In the absence of signals that are required for osteoclast differentiation, 14-3-3 promotes cytosolic localization of MITF, thus regulating its activity [96]. 

The deficiency of the component of activator protein-1 (AP-1), transcription factor c-Fos, impairs osteoclast development and positively regulates osteoclastogenesis [97]. The downregulation of c-Fos using siRNA in OVX mice improved the thickness of the trabeculae, amplified trabecular number, and reduced osteolysis. Additionally, the interleukin (IL) -7/interleukin-7 receptor has been demonstrated to activate the c-Fos/c-Jun pathway to promote RANKL-mediated osteoclast formation, bone resorption, and bone loss in OVX mice [98]. The transcription factor MADS-box transcription enhancer factor 2, polypeptide C (MEF2C) positively regulates osteoclast differentiation by binding to Fos regulatory regions to encourage c-Fos expression, which activates NFATc1 and downstream osteoclastogenesis. Moreover, the deletion of MEF2C in mice increased bone mass and arrested bone erosion due to reduced osteoclast formation [99]. AP-1, MITF, and NFATc1 stimulate cathepsin K (CTSK) promoter activity independent of each other but a combination of any two has a higher positive effect on CTSK promoter activity. However, the combination of AP-1 (c-fos/c-jun) and NFATc1 leads to the largest expression [100].

#### 3.3.1. Cathepsin K

Cathepsins are lysosomal cysteine proteases, and based on their structures, catalytic mechanisms, and target proteins, there are 11 members of cathepsins (cathepsin B, C, F, H, K, L, O, S, V, W, and Z). CTSK is primarily secreted by activated osteoclasts and also by osteoblasts and osteocytes [101]. CTSK is largely responsible degradation of collagen and other matrix proteins during bone resorption and its collagenolytic activity is higher towards matured bone collagen [102]. The study has shown that the deletion of CTSK not only reduces bone resorption, but also improves bone formation. The pharmacological inhibition or deletion of cathepsin K (Ctsk^−/−^ mice) also stimulates bone formation. Cathepsin K directly targets periostin for degradation and deletion of cathepsin K increases periostin and *β*-catenin expression in vivo to improve cortical bone formation. However, periostin deletion selectively inhibits cortical bone formation but not trabecular bone formation in Ctsk^−/−^ mice [103]. CTSK deletion in osteoclasts increases sphingosine kinase 1 (Sphk1) expression, which is responsible for the synthesis of sphingosine 1-phosphate (S1P) that forms a negative feedback loop during osteoclast differentiation. However, high levels of S1P in Ctsk-deficient osteoclasts enhance alkaline phosphatase and mineralized nodules formation in osteoblast cultures in vitro. Additionally, osteoblasts from CTSK-deficient osteoclasts mice exhibit a high RANKL/OPG ratio forming a positive feedback loop to increase the number of osteoclasts [104]. Walia et al. have also demonstrated that CTSK deletion increases osteoblast differentiation by stimulating osteoclast precursor- and osteoclast-mediated secretion of platelet-derived growth factor (PDGF)-BB and S1P. In Ctsk^−/−^ mice, the number of periosteal osteoclast precursors was low under homeostatic conditions; while, after fracture, their number increased, and in fractured Ctsk^−/−^ mice, greater expression of PDGF-BB was observed and contributed to fracture healing [105]. CTSK KO mice also showed enhanced callus mineralization, improved bone healing and remodeling, as well as improved mechanical strength [106].

#### 3.3.2. Osteopontin

OPN belongs to the SIBLING family of cell-matrix proteins and contributes to bone metabolism and homeostasis by regulating adhesion, proliferation, and migration of bone marrow mesenchymal stem cells, osteoclasts, and osteoblasts. Therefore, it has been demonstrated to have a close relationships with the occurrence and development of various bone disorders, such as osteosarcoma, rheumatoid arthritis, and also osteoporosis [107]. Osteopontin acts by influencing the secretion levels of IL-10, IL-12, IL-3, interferon-γ, integrin αv*β*3, nuclear factor kappa B (NF-kB), macrophage, and T cells, and affecting CD44 receptors [108]. Human osteoclasts release OPN into the resorption pit, which acts as a chemokine for succeeding bone formation. Both, α2,3- and α2,6-linked sialic acid play a role in osteoclast differentiation and desialylation can affect osteoclast differentiation in bone [109]. OPN promotes osteoclastogenesis and osteoclast activity via CD44- and αv*β*3-mediated cell signaling and has a decisive role in bone remodeling by contributing to the formation of podosomes, osteoclast survival, and osteoclast motility [110]. However, the addition of exogenous OPN has been shown to inhibit macrophage-to-osteoclast differentiation [111].

#### 3.3.3. Phosphatidylinositol 3-Kinase (PI3K)/Akt Pathway

PI3K/AKT facilitates the anti-apoptotic function in a variety of cell types, including osteoclasts and promotes survival and differentiation. Akt regulates glycogen synthase kinase-3*β* (GSK-3*β*)/NFATc1 signaling to induce osteoclast differentiation. While inhibition of the PI3K reduces osteoclasts formation and NFATc1 expression, Akt overexpression in BMMs favorably stimulates NFATc1 expression. But both of these events do not affect c-Fos expression, which suggests that during RANKL-induced osteoclastogenesis NFATc1 induction mediated by PI3K/Akt is not dependent on c-Fos. Moreover, the overexpression of Akt increases the formation of an inactive form of GSK-3*β* and NFATc1 nuclear localization, and overexpression of the active form of GSK-3*β* decreases osteoclast formation by downregulating NFATc1 [112].

Guanine nucleotide-binding protein subunit α13 (Gα13) belonging to the G protein superfamily controls cell cytoskeleton organization by regulating the RhoGEF-RhoGTPase signaling pathway and negatively regulates osteoclast formation and activity. Osteoclast-specific Gna13 conditional knockout mice have shown severe osteoporosis. Deficiency of Gna13 strongly enhances osteoclast number as well as activity by reducing RhoA activity and enhancing Akt/GSK-3*β*/NFATc1 signaling. Moreover, the inhibition of Akt or RhoA activation rescues the over-activation of Gna13-deficient osteoclasts. Gα13 gain-of-function has an opposite effect on Akt activation and osteoclastogenesis, and produces a protective effect in mice by reducing pathological bone loss in disease models [113]. Moreover, a genome-wide linkage study has identified ARHGEF3 as a strong positional candidate for BMD in women. The product of the ARHGEF3 gene encoding Rho guanine-nucleotide exchange factor (RhoGEF) 3 activates two members of the RhoGTPase family: RHOA, which is required for osteoclast motility and attachment in bone resorption and RHOB, plays a role in bone role in osteoarthritis. The study showed significant associations between single-nucleotide polymorphisms (SNPs) in ARHGEF3 and decreased BMD in post-menopausal women, suggesting the role of the RhoGTPase-RhoGEF pathway in osteoporosis [114].

Conversely, microtubule actin crosslinking factor 1 (MACF1), a spectraplakin protein implicated in cytoskeletal distribution, cell migration, cell survival, and cell differentiation acts as a positive regulator of osteoclast differentiation through Akt/GSK-3*β*/NFATc1 signaling. In primary BMMs derived from osteoporotic mice, MACF1 expression was increased. Though knockdown of MACF1 did not affect the survival of pre-osteoclasts and mature osteoclasts, it diminished Akt and GSK-3*β* phosphorylation and reduced the expression of NFATc1, and inhibited the formation of multinucleated osteoclasts. Furthermore, MACF1 knockdown interrupted actin ring formation in osteoclasts and reduced the area and depth of pits, thus blocking the bone resorption activity of osteoclasts [115]. PI3K/Akt signaling is negatively regulated by phosphatase and tensin homolog (PTEN). Silencing of PTEN expression leads to enhanced Akt and GSK-3*β* phosphorylation levels by RANKL and promotes osteoclasts formation. Inactivation of GSK-3*β* through Akt blocks PTEN phosphorylation and phosphatase activity and improves Akt phosphorylation leading to enhanced osteoclast differentiation [116].

### 3.4. Estrogen and Androgen

Estradiol (E2) in women is produced mainly in ovarian follicles and men, in testes, and from peripheral aromatization. Testosterone is the primary circulating androgen synthesized in the testicles. Estrogens and androgens exert their bone-protective effect through estrogen receptor (ER) α and *β* and the androgen receptor (AR), respectively. These sex-steroid receptor proteins form homodimers and bind to hormone response elements (EREs or AREs), which are DNA sequences [117]. Estrogen affects osteocytes, osteoclasts, and osteoblasts and inhibits bone remodeling, reduces bone resorption, and maintains bone formation, respectively, but the main consequence of estrogen loss is amplified bone resorption [118]. Estrogen is an important key regulator of bone metabolism in men as well as women. The loss of estrogen in menopausal women is accompanied by reduced BMD and increased hip, and vertebral fractures in older women and men [119]. Estrogen controls bone turnover by targeting the expression of RANKL in bone lining cells [120]. The study has shown that the estrogen deficiency in the OVX rat model increases the vascular porosity of cortical bone, decreases bone volume fraction and trabecular number, and enhances trabecular separation in the proximal tibia, diminishing bone strength [121].

The proinflammatory cytokines, such as IL-1, IL-6, TNF-α, GM-CSF, M-CSF, and prostaglandin-E2 (PGE2), increases the number of pre-osteoclasts in the bone marrow and consequently bone resorption. The estrogen downregulates their levels and protects bone from excessive resorption. Estrogen also upregulates TGF-*β*, which acts on osteoclast and decreases their activity, and increases apoptosis [122]. Selective deletion of ERα in differentiated osteoclasts in female mice causes trabecular bone loss but not in males. In wild-type mice, estrogen prompted apoptosis by upregulating Fas ligand in osteoclasts of the trabecular bones but not in ERα deleted osteoclasts mice [123].

The targeted deletion of ERα in mice in mesenchymal osteoblast progenitors and osteoblast precursors reduces periosteal bone apposition and cortical bone mass, whereas, in mature osteoblasts, it has negligible or no effect on bone mass and architecture [117]. Though mice with global deletion of ERα (ERαKO) present confounding systemic effects, mice with tissue-specific ERα deletion in osteoblasts, osteocytes, chondrocytes, or osteoclasts lack the systemic effects. Osteoblast-specific ERαKO female mice (pOC-ERαKO) showed a reduced cancellous bone mass and decreased cortical bone mass. Moreover, osteoblast activity was also reduced in the cancellous bone of the proximal tibia and lowered whole bone strength in the femora and vertebrae [124]. Conversely, conditional deletion of ER*β* specifically in early osteoprogenitor cells (ER*β*-CKO) in the mouse model enhanced the trabecular bone volume fraction. Still, it did not affect cortical bone or bone formation and resorption indices. However, the number of colony-forming unit-osteoblasts (CFU-OBs) was higher in bone marrow cultures from ER*β*-CKO. Quantitative polymerase chain reaction evaluation showed downregulation of 11 out of 16 prespecified pathways in ER*β*-CKO mice [125].

ERα also plays an important role in osteocytes for the formation of trabecular bone in male mice. Male mice lacking ERα protein specifically in osteocytes demonstrated a substantial reduction in trabecular bone volume and reduced rate of bone formation without affecting the number of osteoclasts per bone surface and cortical bone. Mice had reduced expression of RUNX2, Sp7, and DMP1 markers in bone [126]. Female mice lacking ERα in osteocytes (ERα (ΔOcy/ΔOcy)) had a significant reduction in trabecular bone mineral density and bone formation parameters. The potential targets of ERα identified by gene expression microarray and gene ontology analysis were revealed to be Mdk and Sostdc1, the Wnt inhibitors, both of which were highly expressed in osteocytes of ERα (ΔOcy/ΔOcy) mice. Besides, unloading experiments displayed more trabecular bone loss without cortical bone loss [127]. Estrogen deficiency also reduces focal adhesion area and αv*β*3 localization at focal adhesion sites, which increases RANKL/OPG ratio in osteocytes [128].

Deletion of androgen receptor (ARKO) in male mice showed lower serum testosterone levels and reduced cancellous bone volume [129]. Moreover, the inactivation of AR in female mice does not affect bone loss. ARKO mice exhibited high bone turnover with amplified bone resorption, which reduced trabecular and cortical bone mass. Osteoblasts and osteoclasts analysis from ARKO mice has shown that AR was essential for the inhibitory effect of androgens on osteoclastogenesis. RANKL expression was also upregulated in osteoblasts from AR-deficient mice [130]. However, osteoclast-specific ARKO mice did not affect osteoclast surface or bone microarchitecture, suggesting AR in osteoclasts is dispensable for bone maintenance [131]. Male mice with targeted deletion of an AR or ERα in the mesenchymal or myeloid cell lineage displayed reduced bone volume and trabecular number, and the high osteoclast number in the cancellous compartment but no loss of cortical bone mass. This suggests that effect androgens on cortical bone do not require AR or ERα signaling in osteoblasts and osteoclasts [132]. However, AR in osteocytes is essential to maintain male skeletal integrity [133].

## 4. Osteoporosis Treatments

Osteoporosis, a highly prevalent bone disorder worldwide, is characterized by low BMD and an increased risk of osteoporotic fractures [134]. It has been predicted to become a global challenge impacting the quality of life of the affected individuals. Therefore, several treatment options such as medications are available to control disease progression in post-menopausal women and elderly men. Recent studies on the molecular mechanisms in bone have identified novel therapeutic targets for osteoporosis [135].

The proper maintenance of a healthy and active lifestyle is important for bone homeostasis. Physical exercise, a balanced diet, and calcium supplementation improve the quality of osteoporosis patients. Calcium supplementation decreases the rate of bone mineral density and, in combination with vitamin D, reduces the risk of total fractures and hip fractures. However, these methods are inadequate in preventing the progression of osteoporosis. Therefore, pharmacological therapies based on molecular targets have been developed to restore the normal bone balance. The most common therapies are the anti-resorptive and anabolic therapies [136] (Figure 4).

### 4.1. Antiresorptive Drugs

Antiresorptive drugs are aimed towards bone resorption and target osteoclasts. These include bisphosphonates, estrogen and selective estrogen receptor modulators (SERMs), and monoclonal antibodies, such as denosumab and odanacatib.

Bisphosphonates: The bisphosphonates as inhibitors of bone resorption were first identified in the late 1960s and have come to be the important treatment for various skeletal disorders, including osteoporosis [137]. Bisphosphonates bind to calcium exposed areas in the skeleton and cause osteoclast apoptosis. This leads to a reduction in the remodeling rate. Their treatment in hip fracture has shown the ability to increase lifespan [138]. BPs are categorized into two groups: non-nitrogen-containing and nitrogen-containing BPs. Non-nitrogen-containing BPs include clodronate, tiludronate, and etidronate reduces bone resorption through inhibition of ATP-dependent intracellular enzymes by integrating themselves into non-hydrolyzable adenosine triphosphate analogs via class II aminoacyl–transfer RNA synthetases. In comparison, nitrogen-containing BPs, such as alendronate, prevent farnesyl pyrophosphate synthase, which is an indispensable enzyme in the mevalonate pathway essential for the synthesis of cholesterol and isoprenoid compounds [92]. However, more than eight weeks of exposure to bisphosphonates to the bone in the jaw area is linked to a serious adverse event known as bisphosphonate-related osteonecrosis of the jaw (BRONJ) [139].

Estrogen and SERMs: Considering the important role of estrogen in women, deficiency of estrogen in menopausal women predisposes them to declined BMD and increased risks of fractures. Hormonal therapy (HT) has been shown to reduce the incidence of all osteoporosis-related fractures in post-menopausal women [140]. SERMs constitute a group of nonsteroidal compounds that function as ligands for ERs. But, contrasting estrogens that function as ER agonists, SERMs selectively act as agonists or antagonists in a target gene- and tissue-specific manner. SERM express tissue-specific ER subtypes, co-regulatory proteins in various tissues, and varies ER conformational changes by inducing ligand binding. Toremifene, Droloxifene, Idoxifene, and Raloxifene act as agonists and have anti-resoptive effect on bone [141]. The effect of SERMs on the ER downregulates osteoclast activity through TGF-*β*3 and reduces bone resorption [142].

Denosumab: Denosumab targets RANKL and blocks its interaction with RANK. This leads to inhibition in RANKL/RANK signaling and subsequently development and activity of osteoclasts, which reduces bone resorption [143]. The treatment with denosumab in post-menopausal osteoporosis women reduces the risk of vertebral, non-vertebral, and hip fractures and improves BMD [144].

Odanacatib: Odanacatib is a selective and reversible inhibitor of cathepsin K. Osteoclast-mediated bone resorption involves demineralization of inorganic bone mineral and degradation of organic bone matrix. CATK is a lysosomal cysteine protease and has strong collagenolytic activity. It degrades collagen type I and other bone matrix proteins [145]. Thus, cathepsin K inhibition reduces bone resorption without affecting bone formation and increases bone mineral density in post-menopausal women. Nevertheless, odanacatib was linked to an enhanced risk of cardiovascular events, especially stroke, in post-menopausal women with osteoporosis. Therefore, the development of odanacatib for treatment of osteoporosis was no longer pursued [146].

Strontium ranelate: Strontium ranelate consist of two cation atoms of stable strontium, which are closely related to calcium, and a ranelic acid. It has been proposed that strontium ranelate has a dual effect on bone in vitro, but in vivo in OVX rats and monkeys, it has been contradicted. However, in human studies, strontium ranelate showed a dual effect on bone; improved bone formation marker levels and decreased bone resorption markers [147]. The mechanism of action of strontium ranelate is not quite clear but is believed that strontium ranelate inhibits osteoclast function and osteoblast activity via a calcium-sensing receptor (CaSR) by strontium resulting in increased BMD and reduced fracture risks [148].

Calcitonin: Calcitonin is a polypeptide hormone that regulates calcium metabolism in the body and improves bone mineral density and reduces the fracture rate [149]. Calcitonin reduces calcium levels in the systemic circulation through the inhibition of bone resorption or prevention of calcium release from the bone. Therefore, calcitonin is used for the treatment of various bone disorders, including osteoporosis [150]. Salmon calcitonin is analogous to human calcitonin and is used in the treatment of post-menopausal osteoporosis. The treatment of post-menopausal osteoporosis with acupoint injection of salmon calcitonin showed stimulated osteoblast activity and reduced osteoclast-mediated bone absorption [151].

### 4.2. Anabolic Drugs

The anabolic drugs work by enhancing bone formation and include PTH and PTH-related peptides (PTHrP).

PTH and PTHrP: PTH is primarily secreted by the parathyroid glands and enhances trabecular and cortical bone formation, but in hypocalcemic conditions, it increases renal calcium reabsorption and activates calcium from cortical bone by enhancing turnover [152]. PTH or PTHrP can be administered daily and weekly to osteoporotic patients to improve bone mass and lower bone fracture risk. The continuous PTH treatment has catabolic effects on bone, whereas the intermittent PTH administration has anabolic action in osteoporotic patients, and stimulates the proliferation of pre-osteoblasts and osteoblast-mediated bone formation [153]. PTH and PTHrP facilitate osteoblast differentiation and proliferation through extracellular signal-related kinases, cyclin-dependent kinase inhibitor ERK, P27, and RANKL/OPG. Intermittent PTH acts as a Wnt/*β*-catenin signaling agonist and activates the canonical Wnt pathway and inhibits canonical Wnt antagonist, DKK1 mRNA levels in osteoblasts [154]. The PTH-1 receptor (PTH1R), a G-protein-coupled receptor, is the main receptor through which PTH and PTHrP exercise their action, and is essential for the proliferation and differentiation of mesenchymal cells into osteoblasts. PTH and PTHrP have both pro-and anti-apoptotic effects and depend on the cell status. The study showed when mesenchymal and osteoblastic cells were in the preconfluent stage, PTH and PTHrP improved cell viability, whereas in post-confluent stage had the opposite effect. This dual effect of PTH and PTHrP was attributed to the PKA pathway [155]. PTHrP, such as teriparatide and abaloparatide increases bone formation and reduces vertebral as well as non-vertebral fracture risk. Teriparatide and abaloparatide effectively improve BMD by boosting bone architecture and reduces the fracture risk in the upper arm, wrist, or hip [156]. Teriparatide in post-menopausal osteoporosis women has shown significant improvement in both cortical and cancellous bone by enhancing cancellous bone volume and connectivity, trabecular morphology, and cortical bone thickness [157].

Romosozumab: Romosozumab (Evenity^®^) is a human monoclonal sclerostin antibody that promotes bone formation and inhibits bone resorption and has been approved in the EU and the USA to treat severe osteoporosis [158]. In post-menopausal osteoporosis women subcutaneous romosozumab administration showed a reduction in risk of new vertebral and non-vertebral fractures [159].

## 5. Conclusions

Osteoblasts, osteocytes, and osteoclasts are the important bone cells that regulate bone remodeling. The signaling molecules are secreted by one cell act as stimulators or inhibitors for other cells and determine the fate of bone remodeling and, in turn, bone homeostasis. Some molecules affect bone formation or bone resorption, or both. The disruption in their levels either suppresses or promotes bone formation and/or resorption, which can lead to skeletal disorders, including osteoporosis. The insight into the molecular mechanisms behind bone cell function has helped understand the biology of osteoporosis to develop therapeutics by targeting different factors.

The current osteoporosis treatments involve targeting factors to reduce bone resorption or increase bone formation. Anti-resorptive treatments, such as BPs, estrogen and SERMs, denosumab, and odanacatib, target osteoclasts, whereas anabolic drugs, such as PTH and PTH-related peptide (PTHrP) are aimed towards improving bone formation. However, as the pathophysiology of osteoporosis differs among persons, treatments can produce unwanted side effects. The role of different non-coding RNAs, such as micro-, lnc-, and circ-RNA, in regulating bone cell survival, proliferation, differentiation, and apoptosis, has been extensively summarized and therefore can be employed not only as therapeutic, but also targets to improve bone health [160,161]. Different delivery systems developed for targeted gene delivery in different organs, including bone, has opened new opportunities in gene engineering and the preliminary works on microRNA and SiRNA-based gene delivery systems in bone have shown promising results in improving bone microarchitecture and BMD in osteoporosis mice [162]. However, the targeted delivery of therapeutic in vivo remains a challenging aspect in gene-based therapeutics and therefore, further research is warranted.

## Figures and Tables

**Figure 1 ijms-22-08182-f001:**
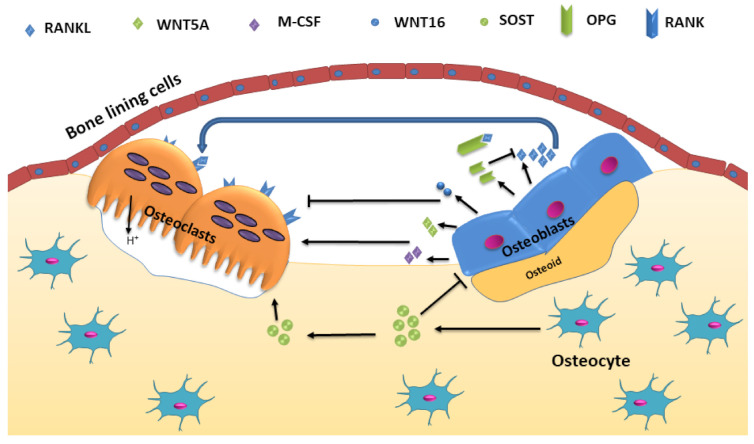
Cellular and molecular interplay during bone remodeling. M-CSF, RANKL produced by osteoblasts positively regulates while WNT5A/16 and OPG negatively regulates osteoclast function. SOST produced by osteocytes embedded in bone matrix positively and negatively regulates the osteoclast and osteoblast functions, respectively.

**Figure 2 ijms-22-08182-f002:**
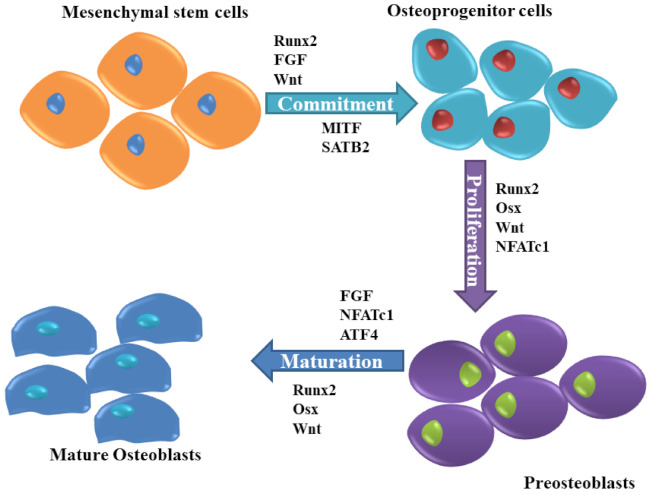
Molecules responsible for osteoblast differentiation. The commitment of mesenchymal stem cells to osteogenic progenitor cells is influenced by Runx2, FGF, Wnt, as well as MITF and SATB2, while Osx, Runx2, NFATc1 promote osteogenic progenitor cell proliferation to preosteoblasts, which is then muture into mature and functional osteoblasts by the action of Runx2, FGF, Wnt, NFATc1, MITF as well as ATF4.

**Figure 3 ijms-22-08182-f003:**
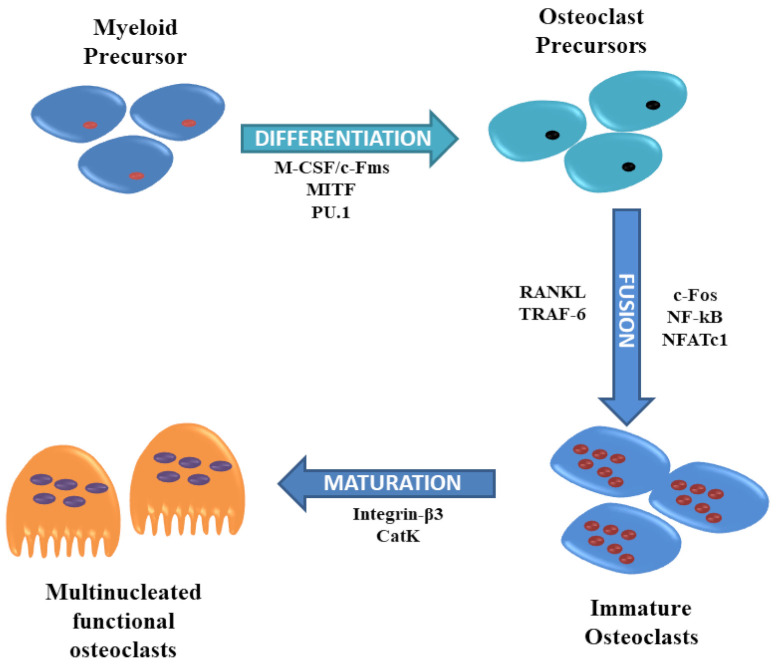
Molecules involved in the process of osteoclast differentiation. M-CSF/c-Fms, MITF, and PU.1 promote differentiation of myeloid precursors into osteoclasts precursor which fuse together to form immature osteoclasts. The maturation of immature osteoclasts to mature osteoclasts is facilitated by integrin-β3 and CatK.

**Figure 4 ijms-22-08182-f004:**
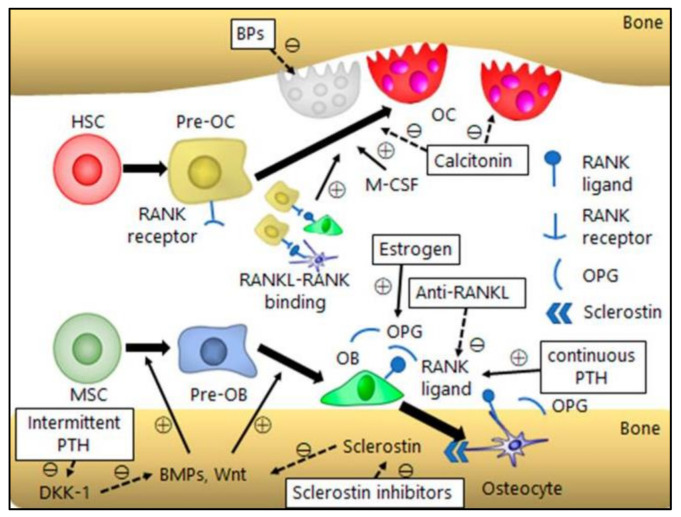
The mechanism of action of osteoporosis therapeutics. The antiresorptive drugs such as bisphosphonate (BP), estrogen, anti-RANKLs (denosumab) and calcitonin; anabolic drugs such as PTH and anti-sclerostins (romosozumab) affecting different bone cells. + mean positive effect; − mean negative effect [10].

## Data Availability

Data sharing not applicable.

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
