# Peer review of "The Development of Molecular Biology of Osteoporosis"

_ijms, 2021, doi:10.3390/ijms22158182_

Round 1

Reviewer 1 Report

In the manuscript "The development of molecular biology of osteoporosis" the authors  focused on 20 molecules produced by osteoblasts, osteocytes, and osteoclasts and their mechanism of action on these cells, and they summarized the different pharmacological osteoporosis treatments. The observations presented by the authors are useful and interesting. All chapters are well written. A manuscript fits into the scope of the journal and could be of interest for the potential readers.

Author Response

Comment: In the manuscript "The development of molecular biology of osteoporosis" the authors focused on 20 molecules produced by osteoblasts, osteocytes, and osteoclasts and their mechanism of action on these cells, and they summarized the different pharmacological osteoporosis treatments. The observations presented by the authors are useful and interesting. All chapters are well written. A manuscript fits into the scope of the journal and could be of interest for the potential readers.

Response: Thank you for your kind response. We appreciate the time and effort provided by you to review the manuscript.

Reviewer 2 Report

This review article discussed the Molecular Biological aspects of osteoporosis. Authors summarized the process and molecules involved in bone remodeling and further therapeutic targets and agents or the treatment of osteoporosis. I think this manuscript is of interest and well written. Overall, this review will be very informative for readers, particularly who are interested in the current osteoporosis treatments involve targeting factors to reduce bone resorption or increase bone formation. There are few concerns that should be addressed.

1. Abstract is redundant. It should be totally revised.

2. The key words “Molecular Biology” and “bone cells” are not suitable in this review. They should be replaced to appropriate words.

3. The subtitle “4. Osteoporosis” should be changed appropriately: e.g., “4. Treatment of osteoporosis”.

4. Authors should add figure legends that explain the figures in detail in all Figures.

5. Figure titles in Figures 2~4 should be revised: e.g., (1) in the Fig. 2, “osteoblast differentiation” instead of “osteoblast activity”, (2) in the Fig. 3, “Molecules involved in the process of osteoclast differentiation”, and (3) in the Fig. 4, “agents for osteoporosis treatment” instead of “agents for osteoporosis”.

Author Response

This review article discussed the Molecular Biological aspects of osteoporosis. Authors summarized the process and molecules involved in bone remodeling and further therapeutic targets and agents or the treatment of osteoporosis. I think this manuscript is of interest and well written. Overall, this review will be very informative for readers, particularly who are interested in the current osteoporosis treatments involve targeting factors to reduce bone resorption or increase bone formation. There are few concerns that should be addressed.

Thank you for your kind response. We appreciate the time and effort provided by you to review the manuscript.

  • Abstract is redundant. It should be totally revised.

Response 1. The abstract has been revised to “Osteoporosis is one of the major bone disorders that affects both women and men and causes bone deterioration and bone strength. Bone remodeling maintains bone mass and mineral homeostasis through the balanced action of osteoblasts and osteoclasts, which are responsible for bone formation and bone resorption, respectively. The imbalance in bone remodeling is known to be the main cause of osteoporosis. The imbalance can be the result of the action of various molecules produced by different one bone cells that acts on other bone cells and influence cell activity. The understanding of the effect of these molecules on bone can help identify new targets and therapeutics to prevent and treat bone disorders. In this article, we have focused on molecules that are produced by osteoblasts, osteocytes, and osteoclasts and their mechanism of action on these cells. We have also summarized the different pharmacological osteoporosis treatments that target different molecular aspects of these bone cells to minimize osteoporosis.”

  • The key words “Molecular Biology” and “bone cells” are not suitable in this review. They should be replaced to appropriate words.

Response 2. The key words “Molecular Biology” and “bone cells” are deleted and replaced with osteoblasts; osteocytes; osteoclasts.

  • The subtitle “4. Osteoporosis” should be changed appropriately: e.g., “4. Treatment of osteoporosis”.

Response 3. We appreciate the suggestion and accordingly “4. Osteoporosis” has been changed to “4. Osteoporosis Treatments”.

  • Authors should add figure legends that explain the figures in detail in all Figures.

Response 4. We have added the explanation of all the figures.

Comment 5. Figure titles in Figures 2~4 should be revised: e.g., (1) in the Fig. 2, “osteoblast differentiation” instead of “osteoblast activity”, (2) in the Fig. 3, “Molecules involved in the process of osteoclast differentiation”, and (3) in the Fig. 4, “agents for osteoporosis treatment” instead of “agents for osteoporosis”.

Response 5. Your suggestions have been taken into the consideration and the titles of Figures 2-4 have been revised. Fig. 2, now reads “Molecules responsible for osteoblast differentiation, Fig. 3, “Molecules involved in the process of osteoclast differentiation. Molecules involved in the process of osteoclast differentiation”, and Fig. 4, “The mechanism of action of osteoporosis therapeutic agents”.